# Preventive and Therapeutic Effects of Baicalein, Galangin, and Isorhamnetin in Chronic Liver Diseases: A Narrative Review

**DOI:** 10.3390/molecules30061253

**Published:** 2025-03-11

**Authors:** Giuseppe Guido Maria Scarlata, Ivo Lopez, Maria Luisa Gambardella, Maja Milanović, Nataša Milić, Ludovico Abenavoli

**Affiliations:** 1Department of Health Sciences, University “Magna Graecia”, 88100 Catanzaro, Italy; giuseppeguidomaria.scarlata@unicz.it (G.G.M.S.); ivo.lopez@studenti.unicz.it (I.L.); marialuisa.gambardella@studenti.unicz.it (M.L.G.); 2Department of Pharmacy, Faculty of Medicine Novi Sad, University of Novi Sad, Hajduk Veljkova 3, 21000 Novi Sad, Serbia; maja.milanovic@mf.uns.ac.rs (M.M.); natasa.milic@mf.uns.ac.rs (N.M.)

**Keywords:** flavonoids, prevention, treatment, Mediterranean diet, fatty liver disease

## Abstract

Chronic liver diseases (CLDs), including fatty liver disease, fibrosis, and cirrhosis, pose significant global health challenges due to the limitedness of therapeutic options. Flavonoids, a class of polyphenolic compounds mainly contained in natural sources and in the foods of the Mediterranean diet, have emerged as potential candidates for liver protection due to their anti-inflammatory, antioxidant, and anti-fibrotic properties. Baicalein, derived from *Scutellaria baicalensis*, exhibits hepatoprotective effects by attenuating oxidative stress, inhibiting fibrogenesis, and modulating lipid metabolism. Galangin, a flavonoid derived from *Alpinia officinarum*, has demonstrated anti-inflammatory and anti-fibrotic properties, while isorhamnetin, a methylated flavonoid found in various fruits and herbs, has been shown to possess hepatoprotective qualities, as it reduces oxidative stress and mitigates inflammation in CLDs. This narrative review highlights the preventive and therapeutic potential of these three flavonoids, emphasizing their role as promising agents for managing CLDs and guiding future research.

## 1. Introduction

Chronic liver diseases (CLDs), including conditions such as fatty liver disease (FLD), hepatitis, and cirrhosis, represent a significant global health burden and often progress to hepatocellular carcinoma, the most common form of primary liver cancer [1]. The pathogenesis of CLDs involves complex interactions between metabolic dysfunction, inflammation, oxidative stress, and immune dysregulation, highlighting the need for effective preventive and therapeutic strategies [2]. At the same time, gut dysbiosis plays a key though still not fully explored role in these conditions [3]. Despite advances in medical treatments such as antiviral therapies, lifestyle modifications, and liver transplantation, the limitations of current approaches, including side effects, long waiting lists in hospitals, and incomplete efficacy, necessitate the exploration of alternative strategies [4,5]. Dietary interventions, particularly those aimed at increasing bioactive compounds, have emerged as promising tools for mitigating the progression of CLDs [6]. Among these compounds, polyphenols, naturally occurring phytochemicals with potent antioxidant, anti-inflammatory, and anti-fibrotic properties, have garnered attention for their protective roles against liver injury and carcinogenesis [7]. Indeed, a dietary regimen abundant in fruits, vegetables, and polyphenol-rich foods, such as those contained in the Mediterranean diet, is associated with a reduced risk of liver-related mortality, suggesting their integration into preventive frameworks [8]. While several polyphenols like quercetin, luteolin, myricetin, and kaempferol have been extensively studied, emerging evidence highlights the potential of baicalein, galangin, and isorhamnetin [9,10,11]. The usefulness of these three polyphenols in preventing and treating CLDs might be explained by the presence in their structure of a C2-C3 unsaturated bond combined with a C-4 carbonyl group, as well as the presence of a C-3 hydroxyl group (in galangin and isorhamnetin) and the di- or tri-hydroxy arrangement in the A ring, and these are structural properties commonly present in natural flavonoids which correlate to their antioxidant properties [12] (Figure 1). Specifically, baicalein, a flavone extracted from *Scutellaria baicalensis*, is known for its antioxidant and hepatoprotective effects, particularly in models of liver fibrosis and FLD [13]. Galangin, a flavonoid derived from *Alpinia officinarum* and other sources, has shown promise in modulating lipid metabolism and suppressing liver inflammation, while isorhamnetin, a methylated flavonol present in plants such as *Hippophae rhamnoides* (sea buckthorn) and onions, showed significant anti-inflammatory and anti-proliferative activities, positioning it as a candidate for liver cancer prevention [14,15]. All three flavonoids naturally occur in the form of glycosides (i.e., baicalin, galangin 7-glucoside, galangin 3-rutinoside, galangin 3-rhamnoside, isorhamnetin 3-O-glucoside, isorhamnetin-7-O-glucoside, isorhamnetin-3,5′-O-β-D-diglucoside, isorhamnetin-3,7-O-β-D-diglucoside, luteoside, etc.) and can be found in different sources, such as *Scutellaria baicalensis*, *Alpinia officinarum*, *Opuntia ficus-indica*, *Hippophae rhamnoides*, *Ginkgo biloba*, onions, apples, pears, and berries [16,17,18]. This narrative review aims to summarize the available literature data regarding the preventive and therapeutic potential of baicalein, galangin, and isorhamnetin in the context of CLDs.

## 2. Baicalein

Baicalein (5,6,7-trihydroxyflavone) is a flavone, a subclass of flavonoids, originally extracted from the roots of *Scutellaria baicalensis*. These features contribute to its potent antioxidant, anti-inflammatory, and cytoprotective properties. As a glycosidic flavon, it is poorly soluble in water but dissolves well in organic solvents and remains stable in neutral to acidic environments [19]. Baicalein’s metabolism produces derivatives like baicalin, which has better water solubility, good permeability, and similar pharmacological effects. Based on the experiments conducted in murine models, baicalin appears to be moderately absorbed in the stomach and poorly absorbed in the small intestine and colon, whereas baicalein had an overall higher absorption than baicalin through the whole gastrointestinal tract [20]. However, baicalin, rather than baicalein, is the major component in the systemic circulation following the oral administration of baicalein [11], and its concentration in plasma can be maintained due to its strong propensity to bind to human plasma albumin [21]. Due to the higher polarity, carrier-mediated transport is required for the lipid bilayer distribution, while the multidrug-resistant protein and the breast cancer resistance protein are identified as baicalin transporters [22]. It was found that baicalin metabolism includes methylation, hydrolysis, hydroxylation, methoxylation, as well as glucuronide and sulfate conjugation, and bile is the major agent in the extraction of the glucuronidated form of baicalin [23]. The unique chemical composition of baicalin offers diverse therapeutic applications, particularly in combating oxidative stress and inflammation [24]. Specifically, this flavone showed promising preventive and therapeutic effects in several preclinical models of CLDs. For instance, intraperitoneal administration of baicalin at a dose of 80 mg/kg for 16 weeks effectively reduced the body weight of rats fed a high-fat diet (HFD) designed to induce liver steatosis. This weight reduction was accompanied by decreased levels of cholesterol and insulin, leading to a subsequent reduction in hepatic steatosis. Similarly, in HepG2 cells, baicalin (at concentrations of 5 and 10 μmol/L) increased AMP-activated protein kinase (AMPK) phosphorylation and reduced lipid accumulation caused by high glucose levels. This approach demonstrated baicalin’s dual effectiveness, addressing both metabolic comorbidities and hepatic steatosis [25]. Furthermore, baicalin effectively reduces liver fibrosis in a mouse model of experimental cholestatic liver injury induced by bile duct ligation (BDL). To determine the effect of baicalin on liver fibrosis, mice received one preoperative (2 h), intraperitoneally injected dose of baicalin and, on alternating days, a postoperative, intraperitoneal injection of baicalin (50 mg/kg/day) for an additional 14 days. Baicalin administration improved fibrosis by downregulating profibrotic markers, reducing proinflammatory cytokines, and mitigating oxidative stress and cell death. It enhanced mitochondrial function and activated the nuclear factor erythroid 2-related factor 2 (NRF2) pathway, leading to increased antioxidant defense through heme-oxygenase 1 (HO-1) and glutamate–cysteine ligase expression. Additionally, baicalin inhibited stellate cell activation, a key process in fibrosis development, by modulating biomarkers such as tissue metallopeptidase inhibitor 1 (TIMP1) [26]. Another study revealed that baicalin has anti-fibrotic effects due to its ability to modulate the activation, proliferation, apoptosis, invasion, and migration of hepatic stellate cells (HSCs) induced by platelet-derived growth factor-BB. Wu and co-authors found that baicalin increased the expression of miR-3595, a microRNA that negatively regulates long-chain-fatty-acid-CoA ligase 4. The inhibition of miR-3595 reversed baicalin’s anti-fibrotic effects, suggesting that baicalin’s action is miR-3595-dependent [27]. At the same time, another study showed that baicalein effectively reduced hepatic fat accumulation in both in vitro and in vivo models of nonalcoholic fatty liver disease (NAFLD). Baicalein activates AMPK and inhibits sterol regulatory element-binding protein-1 (SREBP1) cleavage, thereby suppressing SREBP1’s transcriptional activity and fat synthesis in the liver. Additionally, baicalein improves NAFLD by modulating cholesterol levels, enhancing antioxidant activity, and addressing other biochemical abnormalities [28]. As previously reported, the gut microbiota is involved in the pathogenesis of several CLDs. For instance, a recent study showed that baicalein protects HFD mouse models from NAFLD, restoring gut microbial eubiosis and the expression of key hepatocyte genes involved in lipid metabolism, such as *Apoa4*, *Fabp4*, and *Vldlr* [29]. Shi et al. demonstrated that baicalin mitigated nonalcoholic steatohepatitis (NASH) in an HFD mouse model, reducing lipid accumulation, inflammation, and oxidative damage in liver tissues, partly by enhancing NRF2/HO-1 expression and suppressing the NLRP3/Caspase1/GSDMD-mediated pyroptosis pathway. In vitro experiments confirmed its anti-inflammatory effects [30]. Another recent study demonstrated that baicalin magnesium (which consists of two molecules of baicalin and one magnesium ion) significantly alleviated NASH symptoms in HFD-induced rats. Specifically, it reduced lipid deposition, inflammation, oxidative stress, and liver damage, likely by inhibiting the NLR family pyrin domain containing the 3/caspase-1/interleukin-1 beta inflammatory pathway. Notably, baicalin magnesium, owing to its higher solubility, was more effective than equivalent doses of baicalin or magnesium sulfate, which highlights its superior therapeutic potential for NASH [31]. An additional evaluation showed that baicalin significantly alleviated metabolic dysfunction-associated fatty liver disease (MAFLD) in db/db mice by reducing lipid accumulation and hepatocyte apoptosis. Baicalin also decreased proinflammatory biomarkers and enhanced antioxidant enzymes via the activation of the p62-Keap1-Nrf2 signaling pathway. Co-treatment with an Nrf2 inhibitor weakened these protective effects, confirming Nrf2’s central role in baicalin’s mechanism [32]. Furthermore, Gao et al. confirm that baicalin protects against HFD-induced NAFLD in mice by activating the AMPK pathway. This flavone reduced liver damage by modulating the AMPK-mediated inhibition of the SREBP1 and NF-κB pathways while activating the Nrf2 pathway [33]. This study and the aforementioned one performed by Sun et al. showed that both baicalein and baicalin activate the AMPK pathway, inhibit SREBP1, and improve lipid metabolism and antioxidant activity. However, while baicalein primarily regulates cholesterol levels and addresses metabolic abnormalities, baicalin has a broader impact by also suppressing the NF-κB pathway and activating the Nrf2 pathway, thereby reducing inflammation and oxidative stress. This suggests that although both compounds improve NAFLD through similar mechanisms, baicalin may offer more comprehensive liver protection by targeting additional pathways involved in liver damage [28,33]. Finally, Liu et al. revealed that baicalin exerts a multi-targeted therapeutic effect against hepatic fibrosis, closely linked to gut microbiota modulation. Network pharmacology identified 191 hepatic fibrosis-associated targets, with 9 being specific to baicalin, while experimental validation through genome sequencing showed that baicalin treatment significantly improved gut microbial composition, particularly increasing *Lactobacillus* spp. abundance. Baicalein helps restore gut microbial balance, which in turn influences liver lipid metabolism. This likely happens through gut-derived metabolites, changes in short-chain fatty acids, and the modulation of key gut–liver signaling pathways, such as bile acid metabolism and inflammatory cytokines. One key effect is the increase in *Lactobacillus* spp., which enhances bile salt hydrolase activity. This alters the bile acid composition and activates the farnesoid X receptor, a crucial regulator of liver metabolism. Additionally, improved microbiota composition reduces lipopolysaccharide levels, lowering gut permeability and systemic inflammation—both major drivers of NAFLD progression. Beyond microbiota-driven effects, baicalein and baicalin also have direct actions on hepatocytes. They can activate AMPK, regulate key lipid metabolism genes, and reduce oxidative stress. Their metabolites, such as glucuronides, may undergo enterohepatic circulation, further influencing liver function. Moreover, baicalein shows direct anti-fibrotic activity by inhibiting TGF-β/Smad signaling, suggesting that its hepatoprotective effects are not solely dependent on gut microbiota. Overall, both microbiota-dependent and -independent pathways contribute to baicalein’s benefits in NAFLD, but their indirect effects through gut–liver interactions appear to play the dominant role. These findings highlight baicalin’s potential to treat hepatic fibrosis through its multi-target actions and gut microbiota regulation [29,34]. Table 1 summarizes the different studies regarding the preventive and therapeutic effects of baicalein in CLDs.

## 3. Galangin

Galangin is a flavonol that belongs to the class of flavonoids and is found in high concentrations in the rhizomes of plants such as *Alpinia officinarum*. These plants are native to Southeast Asia, and their rhizomes are used for both culinary and medicinal purposes. In addition, galangin is a significant component of propolis, a resinous substance produced by bees [35]. Although there are no data about the bioaviability of galangin, an in silico study suggests its optimal absorptive, distributive, metabolic, excretive, and toxic (ADMET) properties. In the liver, galangin is metabolized into quercetin and kaempferol; then, it is excreted in the feces after a process of glucuronidation along with other metabolites [36,37]. Modern pharmacological studies have shown that galangin has several pharmacological properties, such as its anticlastogenic, anti-inflammatory, and antimicrobial effects [38,39,40]. Wang et al. evaluated the anti-fibrotic activity of galangin in sixty healthy six-week-old male Sprague Dawley rats. The animals were divided into six groups of ten animals each: the normal control group, the model group, the positive control group (colchicine 0.2 mg/kg), and three groups treated with galangin (20 mg/kg, 40 mg/kg, and 80 mg/kg, respectively). The pattern of liver fibrosis in the rat was induced via intraperitoneal injections of CCl_4_. The study showed that galangin exerts a potent hepatoprotective and anti-fibrotic effect. In mice models of CCl_4_-induced liver fibrosis, galangin reduced the levels of the liver enzymes alanine aminotransferase (ALT) and aspartate aminotransferase (AST), indicating an improvement in liver damage, and restored albumin levels. It also reduced oxidative stress, lipid peroxidation, and collagen accumulation by inhibiting HSCs activation and proliferation. Furthermore, galangin blocked transforming growth factor beta 1 (TGF-β1) and alpha-smooth muscle actin (α-SMA) expression, demonstrating antioxidant and anti-fibrotic properties, thus providing a potential preventive treatment for fibrosis and liver cirrhosis [41]. Another study examined galangin’s antitumor properties in several cancer cell lines. The results of the study provided the first scientific evidence that galangin suppresses the PKC/ERK pathway by reducing TPA-induced activation of matrix metalloproteinase (MMP)-2/-9, thereby inhibiting cell migration and invasion. These results suggest that galangin may be an effective ingredient in the prevention of metastasis and the development of agents for cancer treatment [42]. Furthermore, another study conducted by Zhang et al. evaluated the role of galangin in NAFLD and its activity in reducing fatty degeneration of the liver by inducing autophagy in mice with NAFLD, both preventively and therapeutically. C57BL/6J mice were randomly divided into two different groups: a prevention group, which was treated with galangin (100 mg/kg/d) and an HFD, and a treatment group that received galangin after having been fed an HFD. Some mice were treated with an autophagy inhibitor (3-MA) while being fed an HFD and receiving galangin. The results showed that galangin significantly reduced ALT, AST, body weight, triglycerides, cholesterol levels, and liver damage. Furthermore, the inhibition of autophagy with 3-MA reduced the protective effect of galangin against hepatic steatosis [43]. In addition, a recent study analyzed the anti-fibrotic effects of galangin on human LX-2, an immortalized cell line of activated human hepatic stellate cells (HSCs), activated with TGF-β1 to simulate liver fibrosis in vitro. The data showed that galangin inhibited cell proliferation in a dose-dependent manner, while the induction of cell apoptosis was confirmed by a significant increase in the Bax/Bcl-2 ratio. These molecules are critical for the proliferation and survival of HSCs and for the suppression of the expression of collagen I and α-SMA, key biomarkers of fibrosis. Thus, galangin has proven to be a promising candidate for the treatment of liver fibrosis due to its ability to induce apoptosis in HSCs and modulate key molecular pathways [44]. Another recent study underlines the crucial role of gut–liver axis modulation to alleviate alcoholic liver disease (ALD) in male C57BL/6J mice treated with galangin [45]. This study and the aforementioned one performed by Zhang et al. investigate the hepatoprotective effects of galangin in C57BL/6J mice, highlighting its role in liver disease mitigation through different mechanisms. In the NAFLD study, galangin was tested in both a preventive and therapeutic setting, showing significant reductions in ALT, AST, body weight, triglycerides, cholesterol, and liver damage, with its protective effects largely dependent on autophagy induction, as evidenced by the reduced efficacy in the presence of the autophagy inhibitor 3-MA. In contrast, this last study emphasizes the role of the gut–liver axis in mediating galangin’s protective effects, suggesting that its benefits extend beyond direct hepatic metabolism and involve gut microbiota modulation. While both studies confirm the efficacy of galangin in reducing liver damage in different disease contexts, the NAFLD study focuses on metabolic regulation via autophagy, whereas the ALD study highlights gut–liver axis modulation, indicating that galangin exerts its hepatoprotective effects through multiple pathways depending on the underlying disease pathology [43,45]. Table 2 summarizes the different studies regarding the preventive and therapeutic effects of galangin in CLDs.

## 4. Isorhamnetin

Isorhamnetin is an O-methylated flavonoid derived from quercetin (Figure 1). This compound is noted for its stability and significant antioxidant properties. Isorhamnetin occurs naturally in various sources, including vegetables (e.g., water dropwort, onion, and parsley), fruits (e.g., apples, pears, and berries), and herbs (e.g., green tea and rosemary) [15]. The glycosilated form of isorhamnetin that is found in natural sources is realized via the hydrolytic action of the microflora in the digestive tract and is absorbed via passive diffusion. Isorhamnetin belongs to the Biopharmaceutical Classification System II compounds due to its high permeability and low solubility. As for its structural properties, a methoxy group on the B ring is responsible for its increased stability and lipophilicity in comparison with quercetin, which consequently leads to better permeability and bioavailability [46]. Like other aglycones, extensive first-pass metabolism takes place in both the liver and the gut [47]. Yang et al. investigated the therapeutic potential of isorhamnetin in liver fibrosis. The study employed both in vitro and in vivo models. For in vitro experiments, researchers used LX-2 cells, as well as primary HSCs isolated from mice. Fibrogenic activation was stimulated by administering TGF-β1 at a concentration of 1 ng/mL, while isorhamnetin was administered at different doses (25–100 μM) to evaluate dose-dependent effects. For in vivo experiments, six-week-old Institute of Cancer Research mice (n = 6 per group) were used. Liver fibrosis was induced by an intraperitoneal administration of CCl₄ at a dose of 0.5 mg/kg, while the treatment with isorhamnetin was administered orally at doses of 10 or 30 mg/kg, five days per week. The results demonstrated that isorhamnetin significantly reduced markers of liver damage, including transaminases, as well as collagen deposition and inflammatory infiltration in CCl₄-treated mice. Mechanistically, isorhamnetin inhibited TGF-β/Smad signaling by reducing the phosphorylation of Smad2/3 and suppressed the expression of fibrogenic biomarkers such as α-SMA, plasminogen activator inhibitor-1, and collagen type I alpha 1. Additionally, isorhamnetin activated Nrf2, promoting antioxidant action, including HO-1 and glutamate-cysteine ligase; moreover, it restored glutathione levels and decreased oxidative and nitrosative stress, as evidenced by reduced levels of 4-hydroxynonenal and nitrotyrosine. These findings confirm the dual anti-fibrotic and antioxidant activities of isorhamnetin [48]. Furthermore, Ganbold et al. evaluated the hepatoprotective effects of isorhamnetin in a NASH mouse model, in which male C57BL/6J mice were divided into three groups: control, NASH-induced, and NASH+isorhamnetin. NASH was induced by administering an HFD combined with intraperitoneal injections of CCl₄ and T0901317, an agonist of liver X receptor alpha and beta, while isorhamnetin (50 mg/kg) was administered orally during the last 14 days of the experiment. The treatment significantly improved liver histopathology by reducing lipid and collagen accumulation. Isorhamnetin inhibited de novo lipogenesis by downregulating key genes such as SREBP1c and FAS, reduced hepatic triglyceride levels, and suppressed oxidative stress and HSC activation. Similarly, serum levels of liver damage biomarkers and apoptotic cell numbers in the liver were reduced. Additionally, macrophage infiltration in adipose tissue decreased, improving systemic inflammation. These results highlighted the potential of isorhamnetin as a therapeutic agent in the mitigation of NASH [49]. In addition, a study conducted by Liu et al. examined isorhamnetin’s ability to protect against liver fibrosis by analyzing its effects on mouse models with CCl_4_- and BDL-induced fibrosis. The study was conducted using two different mouse models of liver fibrosis: a CCl_4_-induced model and a BDL-induced model. The treatment consisted of the administration of isorhamnetin (10 or 30 mg/kg) for 5 days/week during 2 or 8 weeks depending on the induced fibrosis model. Four study groups for each model were evaluated: an untreated control group, a model group given CCl_4_ or BDL without isorhamnetin, a group that had been administered a 10 mg/kg dosage of isorhamnetin, and a group that had been administered a 30 mg/kg dosage of isorhamnetin. The results showed that isorhamnetin had reduced the serum levels of AST, ALT, and hydroxyproline; improved the liver’s structure by reducing necrosis, inflammation, and collagen deposition; lowered HSC activation biomarkers; increased quiescence markers; and restored the extracellular matrix balance by modulating the expression of MMP-2 and TIMP1 [50]. Recently, La et al. performed an experimental study using HepG2 and BEL-7402 cells induced with FFAs to simulate a cellular model of NAFLD together with HFD male C57BL/6N mice treated with isorhamnetin (5 mg/kg) or simvastatin as a positive control. The study showed that isorhamnetin alleviated NAFLD by reducing lipid accumulation in the liver, improving levels of triglycerides, cholesterol, and liver function indices. In murine models fed with an HFD, isorhamnetin decreased body weight and adipose fat, improved liver structure, and modulated bile acid metabolism [51]. Table 3 summarizes the different studies regarding the preventive and therapeutic effects of isorhamnetin in CLDs.

## 5. Conclusions and Future Perspectives

Baicalein shows multi-faceted hepatoprotective effects, including antioxidative, anti-inflammatory, and anti-fibrotic properties. It modulates key signaling pathways, such as AMPK, NRF2/HO-1, and NLRP3 inflammasome, while restoring gut microbial eubiosis and improving lipid metabolism. Galangin exerts a hepatoprotective and anti-fibrotic effect by modulating autophagy, inhibiting HSC activation, and reducing oxidative stress. Additionally, its anti-cancer potential and ability to suppress tumor metastasis make it a versatile therapeutic option. Similarly, isorhamnetin exhibits potent antioxidative and anti-fibrotic effects by modulating the TGF-β/Smad and NRF2 signaling pathways, reducing oxidative stress, and improving lipid metabolism (Figure 2). The position of hydroxyl groups along with 2,3 unsaturation combined with a 4-carbonyl group considerably influences baicalein’s, galangin’s, and isorhamnetin’s radical-scavenging, anti-inflammatory, and signaling-modulating activities (Figure 1). It is believed that the higher number of hydroxyl groups in the A ring is responsible for baicalein’s strong tendency to scavenge free radicals (both oxygen- and nitrogen-centered). In addition, galangin’s and isorhamnetin’s hepatoprotective effects are strongly related to the presence of a hydroxyl group in position 3 in the C ring [52,53]. These findings suggest that all three compounds target multiple aspects of CLD pathogenesis, making them promising candidates for future therapeutic development. In spite of these encouraging findings, several gaps remain that warrant further investigation. Firstly, while preclinical studies provide evidence of the efficacy of baicalein, galangin, and isorhamnetin, clinical trials are urgently required so as to validate these effects in humans, assessing their safety and tolerability, as are efficacy studies, which are then to be followed by large-scale clinical trials and real-world effectiveness studies. Moreover, while in vitro experiments with isorhamnetin at concentrations ranging from 25 to 100 µM are relevant for reporting, it should be explicitly stated that these concentrations are not attainable in vivo and are therefore unlikely to achieve the same effects observed in laboratory models. The selection of experimental models is crucial but introduces variability. Animal models such as Sprague Dawley rats and C57BL/6 mice in HFD-induced liver disease studies, BDL models for cholestatic fibrosis, and db/db mice for fatty liver disease provide useful insights but differ in metabolic and immune responses, which affects disease progression and drug metabolism. Likewise, in vitro models like HepG2 and LX-2 cells help elucidate lipid metabolism and fibrosis mechanisms but lack systemic interactions. Differences in genetic background, disease susceptibility, dosing regimens, and pharmacokinetics further impact comparability. For these reasons, standardizing experimental conditions, cross-validating findings across models, and incorporating organoid or primary hepatocyte systems could improve reproducibility and translational relevance. Furthermore, many studies suffer from low statistical power due to the limitedness of sample sizes, which limits the robustness of conclusions and contributes to low reproducibility, often due to constraints in funding, ethical considerations, and the complexity of long-term experimental designs. The bioavailability and pharmacokinetics of these compounds require optimization to ensure effective systemic delivery and therapeutic outcomes. Strategies such as nanoparticle-based delivery systems may enhance their clinical utility [54]. Understanding the interplay between these polyphenols and the gut microbiota composition is critical to harness their full therapeutic potential. Further studies should explore the potentially synergistic effects of these compounds when combined with existing therapies or dietary interventions, such as the Mediterranean diet [55,56]. Finally, the long-term safety and potential off-target effects of these compounds need thorough evaluation. In the near future, integrating baicalein, galangin, and isorhamnetin into precision medicine frameworks could offer personalized therapeutic approaches for patients with CLDs. By combining these polyphenols with advanced omics technologies and artificial intelligence-driven analytics, it may be possible to identify new potential biomarkers predictive of treatment response and to optimize therapeutic strategies for patients [57,58]. Additionally, exploring their potential in preventing liver-related carcinogenesis could open new avenues for cancer prevention in high-risk populations [59]. In conclusion, baicalein, galangin, and isorhamnetin represent promising therapeutic agents for the prevention and treatment of CLDs. Continued research in this area could pave the way for the development of novel polyphenol-based therapies, ultimately improving the quality of life for millions of patients worldwide.

## Figures and Tables

**Figure 1 molecules-30-01253-f001:**
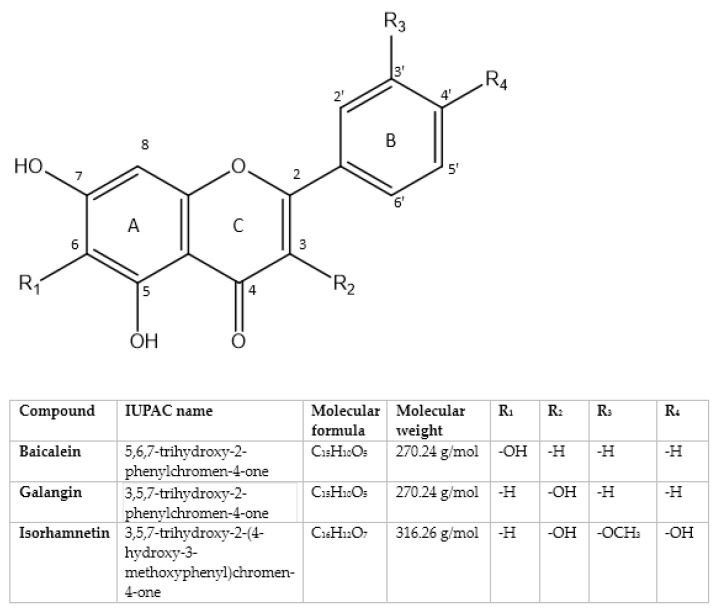
IUPAC representation of baicalein, galangin, and isorhamnetin.

**Figure 2 molecules-30-01253-f002:**
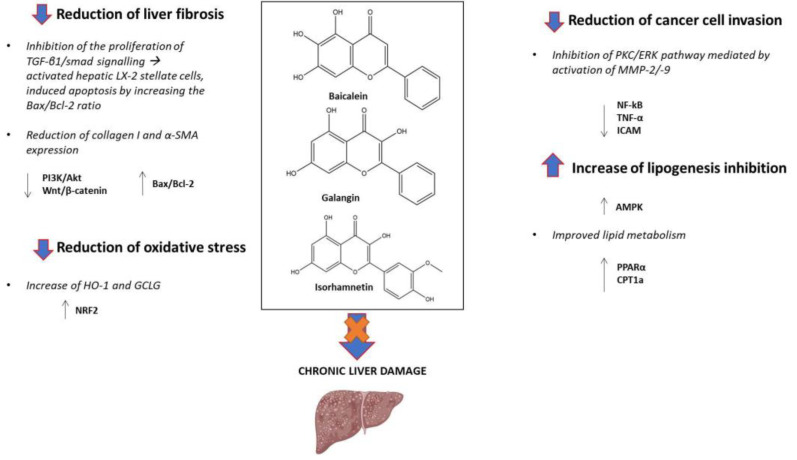
Schematic representation of the preventive and therapeutic effects of baicalein, galangin, and isorhamnetin in CLDs. TGF-β1, transforming growth factor beta-1; Smad, small mothers against decapentaplegic; Bax/Bcl-2, Bcl-2-associated X protein/B-cell lymphoma-2; α-SMA, alpha-smooth muscle actin; HO-1, heme-oxygenase -1; NRF2, nuclear factor erythroid 2-related factor 2; PKC, protein kinase C; ERK, extracellular signal-regulated kinase; MMP, matrix metalloproteinase; NF-κB, nuclear factor kappa B; TNF-α,tumor necrosis factor-alpha; ICAM, intercellular adhesion molecule 1; AMPK, AMP-activated protein kinase; PPAR-α, peroxisome proliferator-activated receptor alpha; CPT1a, carnitine palmitoyl-transferase1A; up arrow, increased levels; down arrow, decreased levels.

**Table 1 molecules-30-01253-t001:** Summary of the different studies regarding the preventive and therapeutic effects of baicalein in CLDs.

Authors	Study Model	Study Groups	Outcome
Guo et al., 2009 [25]	Male Sprague Dawley rats Human hepatoma HepG2 cells	HFD+baicalin (80 mg/kg) vs. HFD control vs. the STD group	Weight reduction was accompanied by decreased levels of cholesterol and insulin, leading to a subsequent reduction in hepatic steatosis. Baicalin increased AMPK phosphorylation and reduced lipid accumulation caused by high glucose levels.
Shen et al., 2017 [26]	Primary stellate cells from male inbred C57BL/6 (H2b) mice	Baicalin (50 mg/kg/day) vs. BDL+baicalin vs. BDL vs. control	Baicalin administration improved fibrosis by downregulating profibrotic markers, reducing proinflammatory cytokines, and mitigating oxidative stress and cell death.
Wu et al., 2018 [27]	HSC-T6 cells	Baicalin (50, 100, and 150 µM) vs. inhibitor vs. control	Baicalin had an anti-fibrotic effect by modulating the activation, proliferation, apoptosis, invasion, and migration of HSC-T6.
Sun, et al., 2020 [28]	HepG2 cells NAFLD mice	NAFLD mice vs. control mice HepG2 cells (baicalein 1 μM or 5 μM) vs. control cells	Baicalein activated AMPK and inhibited SREBP1 cleavage. It also improved NAFLD by modulating cholesterol levels, enhancing antioxidant activity, and addressing other biochemical abnormalities.
Li et al., 2022 [29]	Male C57BL/6N mice	Control vs. treated with silymarin (200 mg/kg) vs. treated with high concentrations of baicalein (200 mg/kg/day) vs. treated with low concentrations of baicalein (100 mg/kg/day)	Baicalein offered protection from NAFLD in HFD mouse models, restoring the gut microbial eubiosis and the expression of key hepatocyte genes involved in lipid metabolism.
Shi et al., 2022 [30]	8-week-old male C57BL/6 mice HepG2 cells	Placebo control vs. normal diet vs. high dose of baicalein (200 mg/kg,day) vs. low dose of baicalein (50 mg/kg,day)	Baicalein’s in vivo and in vitro anti-inflammatory effects mitigated NASH.
Guan et al., 2023 [31]	Male Sprague Dawley rats	The control group vs. the model group vs. the baicalin magnesium groups (50 mg/kg and 150 mg/kg) vs. the baicalin group (146.4 mg/kg) vs. the MgSO4 group (19.7 mg/kg)	Baicalin magnesium was more effective than equivalent doses of baicalin or magnesium sulfate in reducing lipid deposition, inflammation, oxidative stress, and liver damage typical of NASH.
Liu et al., 2023 [32]	Male db/m mice and db/db mice	The control group vs. baicalin 50mg/kg vs. baicalin 100 mg/kg vs. baicalin 200 mg/kg vs. metformin	Baicalin significantly alleviated MAFLD by reducing lipid accumulation and hepatocyte apoptosis.
Gao et al., 2023 [33]	Eight-week-old male C57BL/6J mice AML-12 mouse hepatocyte cell line	The control group vs. the HFD group vs. the HFD+baicalin group (100 mg/kg body weight/day) vs. the baicalin group (100 mg/kg body weight/day)	Baicalin protected against HFD-induced NAFLD in mice by activating the AMPK pathway.
Liu et al., 2023 [34]	SPF-grade SD male rats	The control group vs. the CCl4 group (3 mL kg^−1^) vs. the baicalin group (25 mg kg^−1^)	Baicalin treatment significantly improved gut microbial composition and hepatic fibrosis.

Legend: HFD, high-fat diet; STD, standard diet; BDL, bile duct ligation; HSC-T6, hepatic stellate cells-T6, NAFLD, nonalcoholic fatty liver disease; AMPK, AMP-activated protein kinase; SREBP1, sterol inhibits sterol regulatory element-binding protein-1; NASH, nonalcoholic steatohepatitis; MAFLD, metabolic dysfunction-associated fatty liver disease; CCL4, carbon tetrachloride.

**Table 2 molecules-30-01253-t002:** Summary of the different studies regarding the preventive and therapeutic effects of galangin in CLDs.

Authors	Study Model	Study Groups	Outcome
Wang et al., 2013 [41]	Healthy six-week-old male Sprague Dawley rats	The normal group vs. the positive control group (colchicine 0.2 mg/kg) vs. three groups treated with galangin (20, 40, and 80 mg/kg)	Galangin reduced oxidative stress and collagen accumulation.
Chien et al., 2015 [42]	Chang liver non-cancerous cells HepG2 Hep3B AGS	Chang liver non-cancerous cells vs. HepG2 and Hep3B (0, 1, 2.5, and 5 μM) vs. AGS	Galangin inhibited the PKC/ERK pathway, reducing the activation of MMP-2/-9 and preventing cell migration and invasion.
Zhang et al., 2020 [43]	C57BL/6J mice	The group treated preemptively with galangin (100 mg/kg/day) and HFD vs. the group treated with galangin (100 mg/kg/day) after an HFD vs. the group treated with galangin (100 mg/kg/day) and 3-MA (30 mg/kg, intraperitoneally, three times per week)	Galangin improved several laboratory parameters and hepatic steatosis.
Xiong et al., 2020 [44]	LX-2 cell line	The control group vs. the experimental group treated with galangin (6, 8, and 10 µg/mL)	Galangin inhibited the proliferation of collagen I and α-SMA, along with inducing their apoptosis and reducing their expression. It also suppressed key processes of liver fibrosis.
Duan et al., 2024 [45]	Male C57BL/6J mice	The normal control group vs. the ethanol group vs. the positive control (bifendate pills 150 mg/kg,) group vs. the galangin 30, 90, 150 mg/kg group	Galangin improved gut microbiota imbalance and reduced oxidative stress and inflammation in the liver.

Legend: α-SMA, alpha-smooth muscle actin; PKC, protein kinase C; ERK, extracellular signal-regulated kinase; MMP, matrix metalloproteinase; HFD, high-fat diet; 3-MA, 3-methyladenine; LX-2, human hepatic stellate cells.

**Table 3 molecules-30-01253-t003:** Summary of the different studies regarding the preventive and therapeutic effects of isorhamnetin in CLDs.

Authors	Study Model	Study Groups	Outcome
Yang et al., 2016 [48]	LX2 and HSCs ICR mice	LX-2 cells and HSCs treated with isorhamnetin (25–100 μM) ICR mice treated with sorhamnetin (10 or 30 mg/kg, 5 days/week)	Isorhamnetin significantly reduced inflammation and collagen deposition.
Ganbold et al., 2019 [49]	C57BL/6J male mice	The control group vs. the NASH group vs. the NASH+isoramnetin 50 mg/kg group	Isorhamnetin inhibited lipogenesis, suppressed oxidative stress and the activation of HSCs, and reduced biomarkers of liver damage, apoptotic cells, and systemic inflammation.
Liu et al., 2019 [50]	CCl4-induced mice model BDL mice model	The control group vs. the model group vs. the isorhamnetin group 10 mg/kg vs. the isorhamnetin group 30 mg/kg	Isorhamnetin improved the liver’s structure by reducing necrosis, inflammation, and collagen deposition. It also inhibited HSC activation, blocked autophagy, and attenuated macrophage recruitment.
La et al., 2024 [51]	HepG2 and BEL-7402 cells Male C57BL/6N mice	Standard diet without treatment vs. the HFD group vs. HFD+isorhamnetin (5 mg/kg/day) vs. HFD+simvastatin (5 mg/kg/day)	Isorhamnetin improved liver morphology and modulated bile acid metabolism.

Legend: ICR, Institute of Cancer Research; HSC, hepatic stellate cells; BDL, bile duct ligation; HFD, high-fat diet.

## Data Availability

No new data were created or analyzed in this study. Data sharing is not applicable to this article.

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
