# Peer review of "Preventive and Therapeutic Effects of Baicalein, Galangin, and Isorhamnetin in Chronic Liver Diseases: A Narrative Review"

_molecules, 2025, doi:10.3390/molecules30061253_

Round 1

Reviewer 1 Report

Comments and Suggestions for Authors

The work is promising and merits publication with revisions addressing these points below.

1 The introduction effectively sets the context but lacks a clearly stated hypothesis or primary objective. Explicitly outlining the study's aims would enhance focus.

2 The use of diverse animal models (e.g., Sprague-Dawley rats, C57BL/6 mice) and cell lines (HepG2, LX-2) across studies may introduce variability. Discussing the rationale for model selection and potential comparability issues is needed.

3 Several in vivo studies (e.g., 6 mice per group in Yang et al., 2016) have small sample sizes. Justification for statistical power and reproducibility should be addressed.

4 While dosage ranges are provided, their translational relevance to human therapeutics is unclear. Include a discussion on pharmacokinetics and bioavailability to bridge preclinical and clinical applicability.

5 The paper emphasizes overlapping pathways (e.g., AMPK, NRF2) but could better differentiate the unique mechanistic contributions of each flavonoid. A comparative analysis in the discussion would strengthen insights.

6 The conclusion highlights therapeutic potential but understates the lack of clinical data. Proposing specific steps for clinical trials (e.g., phase I safety studies) would add depth.

7 While baicalein’s microbiota modulation is noted, similar analyses for galangin and isorhamnetin are absent. Address whether this mechanism is compound-specific or a shared feature.

8 Inconsistent use of terms like NAFLD vs. MAFLD (Table 1) may confuse readers. Standardize terminology according to current guidelines.

9 The schematic oversimplifies complex pathways. Consider adding more molecular intermediates (e.g., Smad proteins for TGF-β) to improve mechanistic clarity.

Comments on the Quality of English Language

none

Author Response

Reviewer#1: The work is promising and merits publication with revisions addressing these points below.

Authors’ response: The authors thank Reviewer#1 for the compliments. We appreciate all the comments and suggestions made by Reviewer#1 since they have considerably improved the original manuscript. We have revised the manuscript to address all concerns. All changes made based on the suggestions of Reviewer#1 were highlighted in the revised version.

Specific comments

  1. The introduction effectively sets the context but lacks a clearly stated hypothesis or primary objective. Explicitly outlining the study's aims would enhance focus.

Authors’ response: The authors appreciate this remark. The primary objective of this review paper has been improved and the study aim has been rewritten:

“ The usefulness of these three polyphenols in preventing and treating CLDs might be explained by the presence in their structure of a C2-C3 unsaturated bond combined with a C-4 carbonyl group, as well as the presence of a C-3 hydroxyl group (in galangin and isorhamnetin) and the di- or tri-hydroxy arrangement in the A ring: these are structural properties commonly present in natural flavonoids which correlate to their antioxidant properties [12](Figure 1).”

….

“All three flavonoids naturally occur in the form of glycosides (i.e. baicalin, galangin 7-glucoside, galangin 3-rutinoside, galangin 3-rhamnoside, isorhamnetin 3-O-glucoside, isorhamnetin-7-O-glucoside, isorhamnetin-3,5’-O-β-D-diglucoside, isorhamnet-in-3,7-O-β-D-diglucoside, luteoside, etc .) and can be found in different plant sources such as Scutellaria baicalensis, Alpinia officinarum, Opuntia ficus-indica, Hippophae rhamnoides, Ginkgo biloba, onions, apples, pears, and berries. This narrative review aims to summarize the available literature data regarding the preventive and therapeutic potential of baicalein, galangin, and isorhamnetin in the context of CLDs.”

  1. The use of diverse animal models (e.g., Sprague-Dawley rats, C57BL/6 mice) and cell lines (HepG2, LX-2) across studies may introduce variability. Discussing the rationale for model selection and potential comparability issues is needed.
  2. Several in vivo studies (e.g., 6 mice per group in Yang et al., 2016) have small sample sizes. Justification for statistical power and reproducibility should be addressed.

Authors’ response to points 2 and 3: Thank you for your kind suggestions. We have addressed these concerns in the Conclusion section.

“The selection of experimental models is crucial but introduces variability. Animal models such as Sprague-Dawley rats and C57BL/6 mice in HFD-induced liver disease studies, BDL models for cholestatic fibrosis, and db/db mice for fatty liver disease provide useful insights but differ in metabolic and immune responses, which affects disease progression and drug metabolism. Likewise, in vitro models like HepG2 and LX-2 cells help elucidate lipid metabolism and fibrosis mechanisms but lack systemic interactions. Differences in genetic background, disease susceptibility, dosing regimens, and pharmacokinetics further impact comparability. For these reasons, standardizing experimental conditions, cross-validating findings across models, and incorporating organoid or primary hepatocyte systems could improve reproducibility and translational relevance. Furthermore, many studies suffer from low statistical power due to the limitedness of sample sizes, which limits the robustness of conclusions and contributes to low reproducibility, often due to constraints in funding, ethical considerations, and the complexity of long-term experimental designs.”

  1. While dosage ranges are provided, their translational relevance to human therapeutics is unclear. Include a discussion on pharmacokinetics and bioavailability to bridge preclinical and clinical applicability.

Authors’ response: The authors appreciate this remark. Abrief explanation of baicalein, galangin, and isorhamnetin’s pharmacokinetics and bioavailability has been inserted in the revised version of the manuscript:

“Based on the experiments conducted in murine models, baicalin appears to be moderately absorbed in the stomach and poorly absorbed in the small intestine and colon, whereas baicalein had an overall higher absorption than baicalin through the whole gastrointestinal tract [20]. However, baicalin, rather than baicalein, is the major compo-nent in the systemic circulation following oral administration of baicalein [11] and its concentration in plasma can be maintained due to its strong propensity to bind to human plasma albumin [21]. Due to the higher polarity, carrier-mediated transport is required for the lipid bilayer distribution,while multidrug-resistant protein and breast cancer re-sistance protein are identified as baicalin transporters [22]. It was found that baicalin metabolism includes methylation, hydrolysis, hydroxylation, methoxylation, as well as glucuronide and sulfate conjugation, and that bile is the major agent in the extraction of the glucuronidated form of baicalin [23].”

“Although there is no data about the bioaviability of galangin, an in silico study suggests its optimal absorptive, distributive, metabolic, excretive and toxic (ADMET) properties. In the liver, galangin is metabolized into quercetin and kaempferol; successively, it is excreted in the feces after a process of glucuronidation along with other metabolites [36,37]. Modern pharmacological studies have shown that galangin has several pharmacological properties…”

“The glycosilated form of isorhamnetin that is found in natural sources is realized via the hydrolytic action of the microflora in the digestive tract and is absorbed via passive diffusion. Isorhamnetin belongs to the Biopharmaceutical Classification Sys-tem II compounds due to its high permeability and low solubility. As for its structural properties, a methoxy group on the B ring is responsible for its increased stability and lipophilicity in comparison with quercetin, which consequently leads to better permeability and bioavailability [46]. Like other aglycones, extensive first-pass metabolism takes place in both the liver and the gut [47].”

  1. The paper emphasizes overlapping pathways (e.g., AMPK, NRF2) but could better differentiate the unique mechanistic contributions of each flavonoid. A comparative analysis in the discussion would strengthen insights.

Authors’ response: Thank you for this suggestion. New lines have been inserted in the conclusion section regarding the specific flavonoid features.

As for its structural properties, a methoxy group on the B ring is responsible for its increased stability and lipophilicity in comparison with quercetin, which consequently leads to better permeability and bioavailability.

….

“The position of hydroxyl groups along with 2,3 unsaturation combined with a 4-carbonyl group considerably influence baicalein, galangin, and isorhamnetin’s radical-scavenging, anti-inflammatory, and signaling-modulating activities (Figure 1). It is believed that the higher number of hydroxyl groups in the A-ring is responsible for baicalein’s strong tendency to scavenge free radicals (both oxygen- and nitrogen-centered). In addition, galangin and isorhamnetin’s hepatoprotective effects are strongly related to the presence of an hydroxyl group in position 3 in the C- ring

  1. The conclusion highlights therapeutic potential but understates the lack of clinical data. Proposing specific steps for clinical trials (e.g., phase I safety studies) would add depth.

Authors’ response: Thank you very much for this comment. The conclusion section has been improved and now contains the following lines:

“Firstly, while preclinical studies provide evidence of the efficacy of baicalein, galangin, and isorhamnetin, clinical trials are urgently required so as to validate these effects in humans, assessing their safety and tolerability, as are efficacy studies, which are then to be followed by large-scale clinical trials and real-world effectiveness studies.”

  1. While baicalein’s microbiota modulation is noted, similar analyses for galangin and isorhamnetin are absent. Address whether this mechanism is compound-specific or a shared feature.

Authors’ response: Thank you for this remark. We have added further information regarding the modulation of the gut microbiota by baicalein and conducted a new literature review. In fact, we have added a new recent article regarding galangin, but there is no data regarding isorhamnetin. We have inserted the following lines:

“This likely happens through gut-derived metabolites, changes in short-chain fatty acids, and modulation of key gut-liver signaling pathways, such as bile acid metabolism and inflammatory cytokines. One key effect is the increase in Lactobacillus spp., which enhances bile salt hydrolase activity. This alters bile acid composition and activates farnesoid X receptor, a crucial regulator of liver metabolism. Additionally, improved microbiota composition reduces lipopolysaccharide levels, lowering gut permeability and systemic inflammation—both major drivers of NAFLD progression. Beyond microbiota-driven effects, baicalein and baicalin also have direct actions on hepatocytes. They can activate AMPK, regulate key lipid metabolism genes, and reduce oxidative stress. Their metabolites, such as glucuronides, may undergo enterohepatic circulation, further influencing liver function. Moreover, baicalein shows direct antifibrotic activity by inhibiting TGF-β/Smad signaling, suggesting that its hepatoprotective effects are not solely de-pendent on gut microbiota. Overall, both microbiota-dependent and independent path-ways contribute to baicalein’s benefits in NAFLD, but its indirect effects through gut-liver interactions appear to play the dominant role. These findings highlight baicalin potential to treat hepatic fibrosis through its multi-target actions and gut microbiota regulation [29, 34].”

“Another recent study underlines the crucial role of gut-liver axis modulation to alleviates alcoholic liver disease in male C57BL/6J mice treated with galangin [45].”

  1. Inconsistent use of terms like NAFLD vs. MAFLD (Table 1) may confuse readers. Standardize terminology according to current guidelines.

Authors’ response: Thank you for your comment. All the articles included in Table 1 predate the application of the new MASLD nomenclature (Rinella ME, Lazarus JV, Ratziu V, et al. A multisociety Delphi consensus statement on new fatty liver disease nomenclature. Hepatology. 2023;78(6):1966-1986. doi:10.1097/HEP.0000000000000520). For this reason, the nomenclatures in place until the end of 2023, i.e. NAFLD and MAFLD, were used.

  1. The schematic oversimplifies complex pathways. Consider adding more molecular intermediates (e.g., Smad proteins for TGF-β) to improve mechanistic clarity.

Authors’ response: Thank you very much for this suggestion. The revised manuscript contains an improved Figure 2 (previously Figure 1).

Reviewer 2 Report

Comments and Suggestions for Authors

Page 2: It is not made clear why baicalein, galangin, and isorhamnetin are specifically selected as the polyphenols that would prevent CLDs. References [9] and [10] don’t mention them at all, whereas reference [11] does mention these specific flavonoids, but only among a list of roughly 20 polyphenols. Even there, the three chosen flavonoids don’t seem to stand out particularly as CLD-preventing compounds.

What do baicalein, galangin, and isorhamnetin have in common that justifies a dedicated review paper?

It may be argued that all three compounds share a C2-C3 double bond (common feature of flavones and flavonols).

I would expect that all three flavonoids would naturally occur as glycosides (i.e. baicalin, galangin 7-glucoside, galangin 3-rutinoside, galangin 3-rhamnoside, isorhamnetin 3-O-glucoside, isorhamnetin-7-O-glucoside, isorhamnetin 3,7-O-diglucoside, luteoside, etc.). Glycosides are the most common form in which flavonoids are present in plants.

Since the review is on dietary interventions, can it be made clear in which form the different flavonoids are present in their natural source?

Bacalin was tested in vitro in HepG2 cultures, and in vivo by intraperitoneal administration to rats (see ref [17]). How was baicalein administered in to mice (ref [18])?

It would be ideal if we can get some insight in the ADME properties of baicalin/baicalein – are the 5‑10 µM concentrations of the flavone that were tested in vitro ever reached in vivo?

Please, give the doses that were tested and that triggered a response for all further in vitro and in vivo experiments.

The gut microbiota modulation is an interesting topic. Though, it seems that the effect of baicalin is indirect (baicalein restores gut microbial eubiosis, which in turn affects lipid metabolism). Can some more detail be provided on the sequence of events here? Is it comparable to a cell signalling cascade, or does baicalin/baicalein, or metabolites thereof, directly affect NAFLD?

Pages2,  4, and 7: The descriptions of the three flavonoids are a bit tedious and distract from the overall narrative of the review. Details like the molecular formulas, the molecular weight, or the formal IUPAC names are not further used in the discussion of their pharmacological properties. In addition, for flavonoids, a simplified nomenclature is endorsed by IUPAC and commonly used ( see https://doi.org/10.1515/pac-2013-0919 ).

I would suggest to add a single figure showing the three flavonoids with their names, simplified IUPAC names, and molecular weight. The recommended carbon numbering may be added in the figure so that any discussion on the position of substituents (-hydroxy groups, or glycosides) can be easily interpreted.

Page 5/6: “According to the Authors, galangin reduced lipid accumulation and inflammation, while it increased hepatocyte autophagy and modulated lipid metabolism”.
This is a bit oddly phrased: surely the Authors’ conclusion is underpinned by data. Or is there justifiable reason to doubt the Authors’ conclusion? It would be better to say: The data showed that galangin reduced…etc. If there is doubt, then make that clear and provide links to data that would counteract the Authors’ conclusion.

Page 7: In vitro tests with isorhamnetin in the concentration range 25-100 µM are worth reporting, but is must be made clear that these concentrations would never be obtained in vivo, and are thus unlikely to explain the effects seen in mice.

Minor corrections

Page

1       Dietary interventions, particularly those aimed at increasing bioactive compounds, have…

2       …a dietary regimen…, is associated with…

Author Response

Reviewer#2

General comments

  1. Page 2: It is not made clear why baicalein, galangin, and isorhamnetin are specifically selected as the polyphenols that would prevent CLDs. References [9] and [10] don’t mention them at all, whereas reference [11] does mention these specific flavonoids, but only among a list of roughly 20 polyphenols. Even there, the three chosen flavonoids don’t seem to stand out particularly as CLD-preventing compounds.

What do baicalein, galangin, and isorhamnetin have in common that justifies a dedicated review paper?

It may be argued that all three compounds share a C2-C3 double bond (common feature of flavones and flavonols).

I would expect that all three flavonoids would naturally occur as glycosides (i.e. baicalin, galangin 7-glucoside, galangin 3-rutinoside, galangin 3-rhamnoside, isorhamnetin 3-O-glucoside, isorhamnetin-7-O-glucoside, isorhamnetin 3,7-O-diglucoside, luteoside, etc.). Glycosides are the most common form in which flavonoids are present in plants.

Since the review is on dietary interventions, can it be made clear in which form the different flavonoids are present in their natural source?

Authors’ response: We appreciate all the comments and suggestions made by Reviewer#2 since they have considerably improved the original manuscript. We have revised the manuscript to address all concerns. All changes made based on the suggestions of Reviewer#2 can be found highlighted in the revised version.

We will do our best to reply to three comments  at the same time, as warranted by their interconnection. As already mentioned in the introduction section, several polyphenols like quercetin, luteolin, myricetin, and kaempferol have been extensively studied. Bearing in mind that all three compounds discussed in this review paper have a C2-C3 unsaturated bond combined with a C-4 carbonyl group, a common feature of the natural flavonoids that correlates with their antioxidant activities, as well as C3 hydroxyl group (galangin and isorhamnetin) and di- and even tri-hydroxy arrangement in the A ring we hypothesize that baicalein, galangin, and isorhamnetin might have beneficial effects in the prevention and treatment of CLDs. We completely agree that all three flavonoids naturally occur in form of glycosides. Based on all abovementioned the introduction section is improved with new inserted lines regarding their most common form in natural sources in the introduction section.

  1. Bacalin was tested in vitro in HepG2 cultures, and in vivo by intraperitoneal administration to rats (see ref [17]). How was baicalein administered in to mice (ref [18])?

Authors’ response: Thank you very much for this remark. The revised version of the manuscript contains a rewritten sentence related to ref [18].

“To determine the effect of baicalin on liver fibrosis, mice received one preoperative (2 hours), intraperitoneally-injected dose of baicalin and, on alternating days, a postoperative, intraperitoneal injection of baicalin (50 mg/kg/day) for additional 14 days.”

  1. It would be ideal if we can get some insight in the ADME properties of baicalin/baicalein – are the 5‑10 µM concentrations of the flavone that were tested in vitro ever reached in vivo?

Authors’ response: Authors appreciate this comment and a brief explanation of ADME properties has been provided in the revised version of the manuscript.

“Based on the experiments conducted in murine models, baicalin appears to be moderately absorbed in the stomach and poorly  absorbed in the small intestine and colon, whereas baicalein had an overall higher absorption than baicalin through the whole gastrointestinal tract [20]. However, baicalin, rather than baicalein, is the major component in the systemic circulation following oral administration of baicalein [11] and its concentration in plasma can be maintained due to its strong propensity to bind to human plasma albumin [21]. Due to the higher polarity, carrier-mediated transport is required for the lipid bilayer distribution,while multidrug-resistant protein and breast cancer resistance protein are identified as baicalin transporters [22]. It was found that baicalin metabolism includes methylation, hydrolysis, hydroxylation, methoxylation, as well as glucuronide and sulfate conjugation, and that bile is the major agent in the extraction of the glucuronidated form of baicalin [23].”

  1. Please, give the doses that were tested and that triggered a response for all further in vitro and in vivo experiments.

Authors’ response: We appreciate this suggestion. All the Tables included in the manuscript have been revised according to your suggestions.

  1. The gut microbiota modulation is an interesting topic. Though, it seems that the effect of baicalin is indirect (baicalein restores gut microbial eubiosis, which in turn affects lipid metabolism). Can some more detail be provided on the sequence of events here? Is it comparable to a cell signalling cascade, or does baicalin/baicalein, or metabolites thereof, directly affect NAFLD?

Authors’ response: Thank you very much for this remark. The revised version of the manuscript contains new lines related to gut microbiota modulation.

“This likely happens through gut-derived metabolites, changes in short-chain fatty acids, and modulation of key gut-liver signaling pathways, such as bile acid metabolism and in-flammatory cytokines. One key effect is the increase in Lactobacillus spp., which enhances bile salt hydrolase activity. This alters bile acid composition and activates farnesoid X receptor, a crucial regulator of liver metabolism. Additionally, improved microbiota composition reduces lipopolysaccharide levels, lowering gut permeability and systemic inflammation—both major drivers of NAFLD progression. Beyond microbiota-driven effects, baicalein and baicalin also have direct actions on hepatocytes. They can activate AMPK, regulate key lipid metabolism genes, and reduce oxidative stress. Their metabo-lites, such as glucuronides, may undergo enterohepatic circulation, further influencing liver function. Moreover, baicalein shows direct antifibrotic activity by inhibiting TGF-β/Smad signaling, suggesting that its hepatoprotective effects are not solely de-pendent on gut microbiota. Overall, both microbiota-dependent and independent path-ways contribute to baicalein’s benefits in NAFLD, but its indirect effects through gut-liver interactions appear to play the dominant role. These findings highlight baicalin potential to treat hepatic fibrosis through its multi-target actions and gut microbiota regulation [29,34].”

“Another recent study underlines the crucial role of gut-liver axis modulation to alleviates alcoholic liver disease in male C57BL/6J mice treated with galangin [45].”

  1. Pages2, 4, and 7: The descriptions of the three flavonoids are a bit tedious and distract from the overall narrative of the review. Details like the molecular formulas, the molecular weight, or the formal IUPAC names are not further used in the discussion of their pharmacological properties. In addition, for flavonoids, a simplified nomenclature is endorsed by IUPAC and commonly used (see https://doi.org/10.1515/pac-2013-0919 ).

I would suggest to add a single figure showing the three flavonoids with their names, simplified IUPAC names, and molecular weight. The recommended carbon numbering may be added in the figure so that any discussion on the position of substituents (-hydroxy groups, or glycosides) can be easily interpreted.

Authors’ response: We appreciate this suggestion and a new figure has beenprovided in the revised version of the manuscript. Please see Figure 1.

  1. Page 5/6: “According to the Authors, galangin reduced lipid accumulation and inflammation, while it increased hepatocyte autophagy and modulated lipid metabolism”.

This is a bit oddly phrased: surely the Authors’ conclusion is underpinned by data. Or is there justifiable reason to doubt the Authors’ conclusion? It would be better to say: The data showed that galangin reduced…etc. If there is doubt, then make that clear and provide links to data that would counteract the Authors’ conclusion.

Authors’ response: We appreciate this remark. We have revised the manuscript according to your suggestion.

  1. Page 7: In vitro tests with isorhamnetin in the concentration range 25-100 µM are worth reporting, but is must be made clear that these concentrations would never be obtained in vivo, and are thus unlikely to explain the effects seen in mice.

Authors’ response: Thank you for this comment. We have included this sentence in the Conclusion section.

“Moreover, while in vitro experiments with isorhamnetin at concentrations ranging from 25 to 100 µM are relevant for reporting, it should be explicitly stated that these concentrations are not attainable in vivo and are therefore unlikely to achieve the same effects observed in laboratory models.”

Minor corrections

Page 1:  Dietary interventions, particularly those aimed at increasing bioactive compounds, have…

Page 2: …a dietary regimen…, is associated with…

Authors’ response: Thank you for your comment. The text has been revised by an English language Editor.

Reviewer 3 Report

Comments and Suggestions for Authors

In this narrative review by Scarlata et al, the authors examined the preventive and therapeutic effects of three flavonoids—baicalein, galangin, and isorhamnetin—in management of chronic liver diseases (CLDs). The review highlights the hepatoprotective properties of these compounds, emphasizing their anti-inflammatory, antioxidant, and antifibrotic mechanisms. Baicalein targets lipid metabolism and oxidative stress, galangin modulates autophagy and hepatic stellate cells, while isorhamnetin attenuates fibrosis and lipid accumulation. The authors advocate for further research to validate these promising compounds as potential therapeutic agents for CLDs. This review is well-organized and clearly written. However, there are a number of concerns that need to be addressed before this manuscript is in a publishable fashion. Specific comments are as follows:

1) Sections 3 and 4 appear to have fewer related studies compared to Section 2. However, they contain excessive detail when describing individual experiments. It is not necessary to list all treatment groups and precise experimental outcomes. The authors may focus on the key findings of each study, such as whether a dose-dependent effect was observed, how the flavonoid's effectiveness compares to positive controls, and any unique interpretations or implications of the results.

2) The authors may strengthen the discussion by comparing and contrasting studies with similar designs. For example: References 20 and 25 both investigated the effects of baicalein on nonalcoholic fatty liver disease (NAFLD).  Similarly, references 33 and 35 examined galangin’s role in NAFLD and autophagy regulation.

3) A brief explanation of baicalin magnesium would be helpful, particularly for readers unfamiliar with this with baicalin.

4) LX-2 cells are first introduced in Section 3 when discussing galangin, yet they are described again in Section 4 as if they were being mentioned for the first time.

5) The manuscript should clearly differentiate the effects on hepatocytes and hepatic stellate cells (HSCs), particularly in the summary figure (Figure 1). Pathways such as PI3K/Akt, Bax/Bcl-2, and Wnt/β-catenin are specifically involved in HSC apoptosis and fibrosis regulation, not hepatocyte function. Distinguishing these cellular targets would enhance clarity and prevent potential misinterpretation.

6) While the review summarizes the biological effects (e.g., antioxidant, anti-inflammatory, antifibrotic) and downstream gene regulation triggered by these flavonoids, it would benefit from discussion on the chemical properties underlying these actions. The authors could explain how structural features common to flavonoids enable their radical-scavenging, anti-inflammatory, and signaling-modulating activities. This would offer readers a deeper understanding of the structural-functional relationship that underpins their therapeutic potential. 

Author Response

Reviewer#3

General comments

In this narrative review by Scarlata et al, the authors examined the preventive and therapeutic effects of three flavonoids—baicalein, galangin, and isorhamnetin—in management of chronic liver diseases (CLDs). The review highlights the hepatoprotective properties of these compounds, emphasizing their anti-inflammatory, antioxidant, and antifibrotic mechanisms. Baicalein targets lipid metabolism and oxidative stress, galangin modulates autophagy and hepatic stellate cells, while isorhamnetin attenuates fibrosis and lipid accumulation. The authors advocate for further research to validate these promising compounds as potential therapeutic agents for CLDs. This review is well-organized and clearly written. However, there are a number of concerns that need to be addressed before this manuscript is in a publishable fashion.

Authors’ response: We appreciate all the comments and suggestions made by Reviewer#3 since they have considerably improved the original manuscript. We have revised the manuscript to address all concerns. All changes can be found in red in the revised version.

Specific comments

  1. Sections 3 and 4 appear to have fewer related studies compared to Section 2. However, they contain excessive detail when describing individual experiments. It is not necessary to list all treatment groups and precise experimental outcomes. The authors may focus on the key findings of each study, such as whether a dose-dependent effect was observed, how the flavonoid's effectiveness compares to positive controls, and any unique interpretations or implications of the results.

Authors’ response: Thank you for your positive comment. Tables 2 and 3 have been revised. However, another Reviewer asked to be more specific in presenting the treatment groups.

  1. The authors may strengthen the discussion by comparing and contrasting studies with similar designs. For example: References 20 and 25 both investigated the effects of baicalein on nonalcoholic fatty liver disease (NAFLD). Similarly, references 33 and 35 examined galangin’s role in NAFLD and autophagy regulation.

Authors’ response: Thank you for this suggestion. We have inserted the following lines in the manuscript:

“This study and the aforementioned one performed by Wenlong et al. showed that both baicalein and baicalin activate the AMPK pathway, inhibit SREBP1, and improve lipid metabolism and antioxidant activity. However, while baicalein primarily regulates cholesterol levels and addresses metabolic abnormalities, baicalin has a broader impact by also suppressing the NF-κB pathway and activating the Nrf2 pathway, reducing inflammation and oxidative stress. This suggests that although both compounds improve NAFLD through similar mechanisms, baicalin may offer more comprehensive liver protection by targeting additional pathways involved in liver damage [28,33].”

“This study and the aforementioned one performed by Zhang et al. investigate the hepatoprotective effects of galangin in C57BL/6J mice, highlighting its role in liver disease mitigation through different mechanisms. In the NAFLD study, galangin was tested in both a preventive and therapeutic setting, showing significant reductions in ALT, AST, body weight, triglycerides, cholesterol, and liver damage, with its protective effects largely dependent on autophagy induction, as evidenced by the reduced efficacy in the presence of autophagy inhibitor 3-MA. In contrast, this last study emphasizes the role of the gut-liver axis in mediating galangin’s protective effects, suggesting that its benefits extend beyond direct hepatic metabolism and involve gut microbiota modulation. While both studies confirm the efficacy of galangin in reducing liver damage in different disease contexts, the NAFLD study focuses on metabolic regulation via autophagy, whereas the ALD study highlights gut-liver axis modulation, indicating that galangin exerts its hepatoprotective effects through multiple pathways depending on the underlying disease pathology [43,45].”

  1. A brief explanation of baicalin magnesium would be helpful, particularly for readers unfamiliar with this with baicalin.

Authors’ response: Following your suggestion,e a brief explanation of baicalin magnesium has been added to the revised manuscript:

Another recent study demonstrated that baicalin magnesium (which consists of two molecules of baicalin and one magnesium ion), significantly alleviated NASH symptoms in HFD-induced rats. Specifically, it reduced lipid deposition, inflammation, oxidative stress, and liver damage, likely by inhibiting the NLR family pyrin domain containing the 3/caspase-1/interleukin-1 beta inflammatory pathway. Notably, baicalin magnesium, owing to its higher solubility, was more effective than equivalent doses of baicalin or magnesium sulfate, highlighting its superior therapeutic potential for NASH [23].”

  1. LX-2 cells are first introduced in Section 3 when discussing galangin, yet they are described again in Section 4 as if they were being mentioned for the first time.

Authors’ response: Thank you for this remark. We apologize for the mistake.

  1. The manuscript should clearly differentiate the effects on hepatocytes and hepatic stellate cells (HSCs), particularly in the summary figure (Figure 1). Pathways such as PI3K/Akt, Bax/Bcl-2, and Wnt/β-catenin are specifically involved in HSC apoptosis and fibrosis regulation, not hepatocyte function. Distinguishing these cellular targets would enhance clarity and prevent potential misinterpretation.

Authors’ response: We appreciate this comment and have revised Figure 2. The revised manuscript contains a new Figure 1 following Reviewer#2’s recommendation.

  1. While the review summarizes the biological effects (e.g., antioxidant, anti-inflammatory, antifibrotic) and downstream gene regulation triggered by these flavonoids, it would benefit from discussion on the chemical properties underlying these actions. The authors could explain how structural features common to flavonoids enable their radical-scavenging, anti-inflammatory, and signaling-modulating activities. This would offer readers a deeper understanding of the structural-functional relationship that underpins their therapeutic potential.

Authors’ response: Following your suggestion, a brief explanation related to the structural features common to flavonoids and related to their radical-scavenging, anti-inflammatory, and signaling-modulating activities has been added to the revised version of the manuscript.

As for its structural properties, a methoxy group on the B ring is responsible for its increased stability and lipophilicity in comparison with quercetin, which consequently leads to better permeability and bioavailability.

….

“ The position of hydroxyl groups along with 2,3 unsaturation combined with a 4-carbonyl group considerably influence baicalein, galangin, and isorhamnetin’s radical-scavenging, anti-inflammatory, and signaling-modulating activities (Figure 1). It is believed that the higher number of hydroxyl groups in the A-ring is responsible for baicalein’s strong tendency to scavenge free radicals (both oxygen- and nitrogen-centered). In addition, galangin and isorhamnetin’s hepatoprotective effects are strongly related to the presence of an hydroxyl group in position 3 in the C- ring.

Round 2

Reviewer 1 Report

Comments and Suggestions for Authors

The authors have well revised the manuscript according to my previous comments.

Comments on the Quality of English Language

none

Reviewer 3 Report

Comments and Suggestions for Authors

The reviewer would like to thank the authors for addressing most of the concerns and there are no further questions.